# Hepatic Expression of ACBP Is a Prognostic Marker for Weight Loss After Bariatric Surgery

**DOI:** 10.3390/biom15081173

**Published:** 2025-08-16

**Authors:** Moritz Meyer, Paul Gruber, Christina Plattner, Barbara Enrich, Andreas Zollner, Almina Jukic, Maria Effenberger, Christoph Grander, Herbert Tilg, Felix Grabherr

**Affiliations:** 1Department of Internal Medicine I, Gastroenterology, Hepatology, Endocrinology, and Metabolism, Medical University of Innsbruck, 6020 Innsbruck, Austriapaul.gruber@student.i-med.ac.at (P.G.);; 2Biocenter, Institute of Bioinformatics, Medical University of Innsbruck, 6020 Innsbruck, Austria

**Keywords:** weight loss, ACBP, DBI, bariatric surgery

## Abstract

The incidence and prevalence of obesity and related cardio-metabolic diseases are on the rise, posing a critical health care challenge to systems across the globe. Bariatric surgery is a therapeutic cornerstone for morbidly obese patients, besides novel medical treatments, partly by ameliorating metabolic inflammation, a hallmark of metabolic diseases. Acyl-CoA Binding Protein (ACBP), also known as diazepam-binding inhibitor (DBI), is a regulator of autophagy and metabolism, and has recently been shown to increase in individuals undergoing voluntary fasting and in patients with cancer cachexia-induced malnutrition. By analyzing a prospectively collected study with matched serum and liver samples from patients undergoing laparoscopic adjustable gastric banding at baseline and six months after surgery, we here demonstrate that ACBP serum levels significantly increase following bariatric surgery. Hepatic *ACBP* expression at baseline predicted weight loss six months after the procedure. The predictive value of ACBP warrants further study, as it could identify patients who benefit most from metabolic surgery in the future.

## 1. Introduction

The prevalence of obesity and its associated cardio-metabolic comorbidities—including type 2 diabetes mellitus, cardiovascular diseases, and metabolic dysfunction-associated steatotic liver disease (MASLD)—continues to rise globally, placing an increasing burden on healthcare systems and patients alike [1,2]. A hallmark of these cardio-metabolic diseases is chronic low-grade inflammation, driven by dietary components such as simple sugars and saturated fatty acids [3]. Moreover, cardio-metabolic disorders are frequently associated with gut microbial dysbiosis, which promotes low-grade inflammation in metabolically active tissues, including adipose tissue and hepatic parenchyma [4].

Despite the increasing availability and increasing potency of pharmacological agents to induce weight loss [5], bariatric surgery remains the most effective intervention for morbid obesity. It not only leads to substantial weight reduction but also significantly attenuates inflammation in hepatic and adipose tissues, while improving critical metabolic parameters. Bariatric surgery has been shown to reduce inflammation in hepatic and adipose tissue and to improve key metabolic parameters, such as the Homeostasis Model Assessment of Insulin Resistance (HOMA-IR) [6,7,8]. We have previously demonstrated that bariatric surgery significantly ameliorates liver damage and inflammation in MASLD patients [8,9,10].

Acyl-Coenzyme A Binding Protein (ACBP), encoded by the diazepam binding inhibitor gene (DBI), is an orexigenic protein that demonstrates multifunctional activities [11]. For example, ACBP/DBI is involved in intracellular lipid metabolism and acts as a paracrine signaling molecule by binding to γ-aminobutyric acid receptors (GABA_A_ receptors) [12,13]. It is assumed that ACBP is released from cells in an autophagy-dependent manner [14,15] and ACBP is able to regulate autophagy vice versa. Recently, it was described that extracellular ACBP inhibits autophagy [11,16], which is discussed to be a possible negative feedback loop [11].

In preclinical models, genetic deletion of ACBP, both systemically and in adipose tissue specifically, prevented high-fat diet-induced weight gain [17]. Furthermore, ACBP has been shown to stimulate appetite via GABA_A_ receptor signaling [12], and administration of recombinant ACBP reduces circulating leptin levels [16]. Pharmacological inhibition of ACBP/DBI has also been shown to ameliorate liver injury in murine models [18]. Collectively, these findings highlight ACBP/DBI as a potentially important regulator of metabolic homeostasis.

Clinical data suggest that serum ACBP/DBI levels are decreased in patients with anorexia nervosa, while they are elevated in obese individuals [16]. Notably, bariatric surgery was associated with a decrease in serum ACBP/DBI one year post-intervention, whereas voluntary fasting, cancer-associated undernutrition, and chemotherapy-induced weight loss have been linked to increased circulating ACBP levels, possibly as a counter mechanism to tackle weight loss [17].

However, whether bariatric surgery influences tissue-specific ACBP/DBI expression remains unclear. In this study, we aimed to evaluate the effects of weight loss induced by laparoscopic adjustable gastric banding (LAGB) on ACBP/DBI expression in hepatic and adipose tissue.

## 2. Methods

### 2.1. Study Design

The original study was prospectively carried out between 2003 and 2006. The patient selection was performed at the department of internal medicine at the Medical University of Innsbruck. Morbidly obese patients, with a body mass index (BMI) greater than 35 kg/m^2^, with no other significant medical, physical, or psychosocial disabilities, were included in the study [7,8,9]. Patients with statin treatment, alcohol intake exceeding 20 g per week, or other chronic liver diseases such as Hepatitis B or C infection, hemochromatosis, Wilson’s disease, autoimmune hepatitis, primary sclerosing cholangitis, or primary biliary cholangitis were excluded from the cohort. The study protocol was approved by the ethics committee of the Medical University of Innsbruck (AN 1623 197/4.11), and written informed consent was obtained from all participants prior to undergoing laparoscopic adjustable gastric banding (LAGB) and sample collection. The Hepatic tissue and adipose tissue samples were collected intraoperatively during LAGB and again six months after the surgery by ultrasound-guided planned follow-up biopsy in the same cohort. Blood samples were collected pre-operatively and 6 months after surgery. Included in this analysis were 30 patients of the original study. Biopsy specimens were stored at −80 °C for further analysis. Clinical and laboratory parameters (such as HOMA-IR, insulin levels, CRP, and FIB4) were collected via the routine laboratory and calculated as usual (e.g., FIB4). All human studies were conducted in accordance with the Helsinki Declaration.

### 2.2. Quantification of Hepatic and Adipose Tissue ACBP/DBI Expression

Expression analysis was performed on samples from a subset of patients (*n* = 23; 18 females, 5 males) from the original study, where samples were still available. Total RNA was extracted from thawed tissue samples using TRIzol Reagent (Invitrogen, Carlsbad, CA, USA). cDNA was synthesized via reverse transcription using Moloney Murine Leukemia Virus Reverse Transcriptase (Invitrogen, Carlsbad, CA, USA). Quantitative real-time PCR was conducted using the C1000 Touch system (BioRad, Hercules, CA, USA) with GoTaq^®^ qPCR Master Mix (Promega, Madison, WI, USA). ACBP/DBI expression was normalized to the reference gene *GAPDH*. The following primer sequences were used: *GAPDH*-forward: 5′-GTC GCC AGC CGA GCC-3′; *GAPDH*-reverse: 5′-CCC AAT ACG ACC AAA TCC GT-3′; *ACBP/DBI*-forward: 5′-CAG AGG AGG TTA GGC ACC TTA-3′; *ACBP/DBI*-reverse: 5′-*TAT GTC GCC CAC AGT TGC TTG*-3′

### 2.3. Isolation of Human Peripheral Blood Mononuclear Cells (PBMCs)

Human peripheral blood mononuclear cells (PBMCs) were isolated from the blood of three healthy volunteers who provided informed consent. Blood was collected into heparinized tubes, and density gradient centrifugation was performed using Lymphoprep solution according to the manufacturer’s protocol (Axis Shield, Oslo, Norway). In brief, peripheral blood was diluted 1:1 with PBS + 2% FCS and afterwards carefully layered onto Lymphoprep solution in a centrifuge tube, ensuring minimal mixing at the interface. The samples were centrifuged at 800× *g* for 20 min at RT with the centrifuge brake turned off. After centrifugation, the upper plasma layer was removed, and the mononuclear cell layer located at the interface was carefully aspirated and transferred into a new tube. The isolated PBMCs were then washed with culture medium and centrifuged at 300× *g* for 5 min. The cell pellet was resuspended for subsequent applications.

### 2.4. Cell Culture and Stimulation

Both HepG2 cells and PBMCs were cultured in RPMI medium supplemented with 10% fetal calf serum (Sigma, St. Louis, MO, USA), as previously described [7,8]. After seeding, cells were grown overnight and were then stimulated for 24 h with recombinant human TNF (50 ng/mL) (Peprotech, Waltham, MA, USA), recombinant human IL-1β (1 ng/mL) (Peprotech, Waltham, MA, USA), lipopolysaccharide (LPS, 100 ng/mL) (Invivogen, San Diego, CA, USA), or a vehicle control. After the 24-h stimulation period, cells were harvested, and RNA was extracted for further analysis as described earlier.

### 2.5. Histological Evaluation

The specimens were fixed in formalin, paraffin-embedded, sectioned, and stained with H and E as well as chromotrope-aniline blue (Roques’ trichrome). Histopathological scoring was performed by a professional pathologist on samples of 14 patients (11 females and 3 males) from the original study. We evaluated the grade of steatosis, inflammation, and necrosis as previously described [9]. Briefly, the grade of steatosis was assessed by using a 4-grade classification modified from Kleiner et al. 2005 [19]: 0, absent; 1, <5%; 2, 5–33%; 3, 33–66%; 4, >66%. The grading as well as staging were performed according to Brunt et al. 1999 [20] with: stage 1, zone 3 perisinusoidal fibrosis; Stage 2, zone 3 perisinusoidal fibrosis plus focal or extensive portal fibrosis; Stage 3, zone 3 perisinusoidal fibrosis with bridging fibrosis; and Stage 4, cirrhosis. The NAFLD activity score (NAS) was calculated according to Kleiner et al. 2005 [19].

### 2.6. Determination of Serum ACBP/DBI Levels

Serum levels of ACBP/DBI were analyzed using Acyl-CoA-binding Protein (human) ELISA Kit, Cat. No. AG-45B-0019-KI01 (AdipoGen, San Diego, CA, USA), according to the manufacturer’s protocol, in a subset of 30 patients from the original study, one sample was excluded as an outlier after statistical testing. Briefly, 100 µL of serially diluted standards as well as 100 µL of serum were added in duplicate to the wells of a pre-coated microtiter plate. The plate was incubated for 2 h at RT. Following incubation, the wells were aspirated and washed 5 times with 300 µL of the wash buffer. Subsequently, 100 µL of the diluted detection antibody was added to each well. Afterwards, the plate was incubated for 1 h at RT. Wells were then washed 5 times as described above. Next, 100 µL of diluted HRP-labeled streptavidin was added to each well and incubated for 30 min at RT. After a further series of 5 washes, 100 µL of TMB substrate solution was added to each well, and the plate was incubated in the dark at RT for 20 min to allow color development. The enzymatic reaction was stopped by adding 100 µL of stop solution to each well, resulting in a color change from blue to yellow. Optical density (OD) was measured at 450 nm using a microplate reader.

### 2.7. Statistical Analysis

Normality distribution was assessed using Kolmogorov-Smirnov and Shapiro-Wilk test. Data were expressed as mean ± standard error of mean (SEM). Pairwise comparisons were carried out using paired Student’s *t*-test or Wilcoxon signed-rank test. Regression analysis was calculated as a simple linear regression. Classification of patients into responders or non-responders was carried out using a cut-off value of 20 kg (corresponding to the median weight loss of the entire patient cohort). Outlier detection was performed prior to each statistical analysis using Grubbs outlier detection test. All analyses were performed using GraphPad Prism 10.1.2 (GraphPad Software, Boston, MA, USA). Statistical significance was considered when the *p*-value was ≤0.05.

## 3. Results

### 3.1. Bariatric Surgery-Induced Weight Loss Ameliorates Metabolic Function and Increases ACBP/DBI in Serum

We included 30 patients with morbid obesity (25 females and 5 males) in this study. The cohort had a mean BMI of 42.69 ± 3.54 kg/m^2^ (for detailed patient characteristics, see Table 1). As expected, LAGB led to a significant weight loss in all but one patient after 6 months, with a median weight loss of 20 kg and a median BMI reduction of 15.81%.

This weight loss was accompanied by an improvement in metabolic function, as reflected by decreased HOMA-IR, fasting insulin, and serum leptin levels, along with a reduction in C-reactive protein (CRP) (see Table 1) and other pro-inflammatory cytokines, which was previously reported in this cohort [7,8,9]. Moreover, LAGB induced weight loss significantly reduced the FIB-4 score, a non-invasive index to estimate liver fibrosis, comprising age, ALT, AST, and thrombocyte count.

Interestingly, we also observed a significant increase in serum ACBP/DBI levels at the time point of 6 months post-surgery (Figure 1A), consistent with changes also seen in patients undergoing voluntary fasting-induced weight loss described earlier [12]. This increase was only statistical significant in female patients Appendix A.

However, in paired hepatic and adipose tissue samples, we did not observe an increase in *ACBP/DBI* expression (Figure 1B,C), indicating that the rise in serum ACBP/DBI is likely independent of production in the liver or subcutaneous adipose tissue.

### 3.2. A Lower Expression of Liver ACBP/DBI Is Observed in Those with a Subsequent Augmented Weight Loss

In the next step, we investigated whether ACBP/DBI could serve as a prognostic marker for weight loss following LAGB. Using a cut-off of 20 kg absolute weight loss—the median in our study cohort—we stratified participants into responders (≥20 kg weight loss) and non-responders (<20 kg weight loss). Strikingly, *ACBP/DBI* expression in hepatic tissue at baseline was significantly lower in responders compared to non-responders (Figure 2A), suggesting that hepatic *ACBP/DBI* expression may predict the response to bariatric surgery-induced weight loss.

Interestingly, baseline liver expression of *KLF3* and *METRNL*—two targets previously implicated in LAGB-induced weight loss [7,8]—was positively correlated with hepatic *ACBP/DBI* expression (Figure 2B,C). This suggests that ACBP/DBI may engage in paracrine signaling within the liver, potentially contributing to resistance to weight loss after LAGB.

However, we did not observe significant differences in serum ACBP/DBI levels between responders and non-responders, either at baseline (Figure 2D) or six months post-surgery, indicating that the prognostic value may be specific to hepatic tissue expression rather than circulating levels.

### 3.3. Serum ACBP/DBI Positively Correlates with Reduction of Steatosis and Fibrosis Six Months After LAGB

Weight loss induced by LAGB led to a significant reduction in histological steatosis six months after surgery (Figure 3A), as previously reported for this cohort [9].

To investigate whether baseline levels of ACBP/DBI could predict this response, we correlated serum ACBP/DBI concentrations at the time of surgery (TP0) with the degree of histological steatosis at six months post-surgery (TP6).

Notably, we observed a significant positive correlation between serum ACBP/DBI levels at the time of surgery and steatosis severity at TP6, indicating that higher baseline ACBP/DBI may be associated with a poorer histological response towards LAGB-induced amelioration of MASLD.

In addition, baseline serum ACBP/DBI levels also correlated positively with the degree of histological fibrosis at TP6. However, there was no significant overall change in fibrosis between the time of surgery and TP6 Appendix A.

### 3.4. Serum ACBP/DBI Predicts NAS Six Months After LAGB

To further explore whether ACBP/DBI signaling contributes to the progression of metabolic dysfunction-associated steatotic liver disease (MASLD), we examined the relationship between ACBP/DBI expression and the NAFLD activity score (NAS), a histological index used as a surrogate for the diagnosis of non-alcoholic steatohepatitis (NASH) [19].

In our cohort, serum ACBP/DBI levels at the time of surgery were positively correlated with the NAS at the same time point (Figure 4A). Furthermore, patients with a NAS ≥ 5—often used as a cut-off to define NASH—exhibited significantly higher serum ACBP/DBI levels compared to those with NAS ≤ 4 (Figure 4B). Consistently, serum ACBP/DBI levels at the time of surgery also positively correlated with NAS at TP6 (Figure 4C), suggesting that circulating ACBP/DBI may reflect or contribute to ongoing hepatic inflammation and injury.

Surprisingly, we did not observe a significant correlation between hepatic tissue *ACBP/DBI* expression and NAS at either time point (Figure 4D,E), implying that serum ACBP/DBI levels—but not hepatic expression—are associated with MASLD severity and progression in this cohort.

### 3.5. LPS Stimulation Reduces ACBP/DBI Expression in Human PBMCs

To investigate whether pro-inflammatory stimuli regulate *ACBP/DBI* expression, we stimulated HepG2, a human hepatoma cell line, and peripheral blood mononuclear cells (PBMCs) from healthy human donors with lipopolysaccharide (LPS), IL-1β, and TNFα.

Unexpectedly, none of the pro-inflammatory stimuli led to an increase in *ACBP/DBI* expression in either HepG2 cells or PBMCs. Interestingly, LPS stimulation significantly reduced *ACBP/DBI* expression in PBMCs (Figure 5A), but had no effect in HepG2 cells (Figure 5B). These findings suggest that endotoxemia—a hallmark of metabolic inflammation—may suppress ACBP/DBI in immune cells, potentially contributing to inflammation-associated metabolic dysregulation.

## 4. Discussion

ACBP/DBI is a key mediator of intracellular metabolic pathways, serves as a regulator of paracrine signaling via its interaction with GABA_A_ receptors, and is a potent orexigenic protein [11,12,16]. In this study, we show that serum ACBP/DBI levels are significantly increased six months following weight loss induced by bariatric surgery in our cohort. This finding contrasts with a recent study by Bravo-San Pedro et al. [16], which reported a decrease in ACBP/DBI levels one year after gastric bypass surgery. While our study used gastric banding, where weight loss is mainly induced through restriction of food uptake, the study from Bravo-San Pedro et al. [16] observed the decrease in patients undergoing gastric bypass, where weight loss is induced not only due to restriction but also due to malabsorption and other factors [21]. Thus, the opposite observations could be due to different mechanisms of weight loss in the different modalities of bariatric surgery. However, other studies, including one by Joseph et al., have shown that both tumor-associated malnutrition and voluntary weight loss (without invasive interventions) lead to an increase in serum ACBP/DBI [12,17]. Additionally, Joseph et al. [17] demonstrated that ACBP/DBI levels exhibit a dual regulation pattern: in a long-term follow-up, individuals who experienced weight gain followed by weight loss showed an increase in ACBP/DBI, a pattern also observed in those maintaining stable weight over the same period.

ACBP/DBI has been linked to increased food intake. The upregulation of ACBP/DBI we observed six months post-surgery may therefore represent a counter-regulatory mechanism aimed at maintaining body weight. Thus, it is possible that ACBP/DBI does not directly contribute to weight loss, but given the orexigenic effects of ACBP/DBI, this could reflect a physiological response to maintain homeostasis and counteract weight loss. Weight loss after bariatric surgery is associated with a decrease in pro-inflammatory cytokines and is, per se, associated with an improvement in metabolic function [22]. Paradoxically, extracellular ACBP/DBI inhibits autophagy, and high levels of ACBP/DBI were associated with cardiovascular disease [23]. Thus, it is possible that the increase in serum ACBP/DBI after LAGB would negatively influence metabolic function, but other positive alterations of metabolic health after weight loss outweigh this negative influence. Further studies would be needed to investigate this question.

Interestingly, we observed a positive correlation between *ACBP/DBI* and *KLF3* and *METRNL*, both of which we previously linked to weight loss in LAGB patients. This observation suggests that ACBP/DBI could be involved in endocrine signaling within the liver of morbidly obese individuals, which may contribute to the metabolic adaptation to weight loss.

Obesity-associated systemic low-grade inflammation is well documented, and LPS stimulation of PBMCs from healthy individuals reduced *ACBP/DBI* expression. Endotoxemia, which is a hallmark of systemic inflammation [24,25], is reduced after bariatric surgery [26]. The observed increase of serum ACBP/DBI after LAGB in our cohort could thereby also be a result of reduced systemic inflammation rather than a counteracting mechanism of the body to maintain body weight.

Using the median weight loss after LAGB, we divided our cohort into responders and non-responders. Importantly, this did not mean that non-responders had no weight loss, but rather that responders had a better outcome, using weight loss as an outcome marker. Liver expression of *ACBP/DBI* at the time point of surgery was significantly lower in the responder group, compared to the non-responder group. This observation indicates that the liver expression of *ACBP/DBI* could act as a biomarker to choose the right treatment for the patient, especially if liver biopsy is performed before operation due to other reasons (e.g., evaluation of liver disease). Otherwise, one could speculate that in patients who underwent liver biopsy before bariatric surgery, those with high expression levels could be directed towards other obesity therapies, such as GLP-1 receptor agonists. Furthermore, if ACBP/DBI could act as a biomarker also for long-term outcome, it has to be investigated in additional studies with a longer follow-up period than our current study. A lot of biomarkers have been investigated for their potential as predictors of outcome after bariatric surgery. While, for example, preoperative leptin has been shown to predict glycemic control post-surgery [27], there is currently no biomarker in clinical routine use that helps guide therapy selection in specific patients [28]. However, studies, such as the DECON trial, could help us in the future to detect new biomarkers in patients in need of bariatric surgery [29].

In this and recent studies, we also demonstrate that weight loss induced by LAGB leads to an overall improvement of MASLD, as reflected by a significant reduction in hepatic steatosis. Notably, we also found that higher serum concentrations of ACBP/DBI at baseline were positively correlated with the degree of steatosis and fibrosis six months after surgery. These findings suggest that elevated preoperative ACBP/DBI levels may predict a less favorable histological response to weight loss, further highlighting a potential role for ACBP/DBI as a biomarker for patient stratification. Given the increased knowledge about the role of ACBP/DBI in metabolic regulation and autophagy, it is conceivable that this protein could influence the extent of hepatic fat resolution following bariatric interventions.

Additionally, we observed that serum ACBP/DBI levels at baseline were positively correlated with NAS (Non-Alcoholic Fatty Liver Disease Activity Score) six months post-surgery, suggesting a link between ACBP/DBI levels and liver inflammation. This finding aligns with recent studies, which have shown correlations between plasma ACBP/DBI, NAFLD, and FIB4, as well as with other evidence, which demonstrates that ACBP/DBI inhibition ameliorates liver inflammation in murine models [30].

The observation that liver expression of *ACBP/DBI* in weight loss responders was lower than in non-responders, despite no significant difference in serum ACBP/DBI levels, implies that the hunger-stimulatory effects of ACBP/DBI are not the sole mechanism influencing weight loss outcomes after LAGB [11].

## 5. Conclusions

In summary, this study demonstrates that serum ACBP/DBI levels significantly rise following bariatric surgery, while hepatic *ACBP/DBI* expression at baseline predicts postoperative weight loss. Moreover, patients with higher ACBP/DBI serum levels at baseline showed a less favorable response towards the amelioration of MASLD six months after LAGB. These data underscore a potential role for ACBP/DBI as a potent biomarker to identify patients who would benefit most from bariatric interventions by LAGB. While the liver expression of ACBP/DBI can only be assessed by invasive liver biopsy, which is usually not needed for the diagnosis of MASLD [31], serum levels of ACBP/DBI can be easily assessed. Future prospective studies now need to clarify if ACBP/DBI liver expression and serum levels indeed predict response towards bariatric surgery. Additionally, further studies are now warranted to elucidate the mechanistic links between ACBP/DBI and postoperative weight loss. Such studies could also be used to define additional, new therapeutic targets in tackling weight disorders and associated metabolic diseases. As less invasive therapies, such as GLP-1 receptor agonists, are increasingly used worldwide to treat obesity and associated diseases, it would also be of interest if ACBP/DBI might act in this setting as a potential biomarker. In conclusion, our findings suggest that ACBP/DBI may serve as a potential biomarker for predicting weight loss outcomes following bariatric surgery. If this observation is validated in independent cohorts, ACBP/DBI could contribute to the development of personalized therapeutic strategies for obesity and associated metabolic disorders in the future.

## Figures and Tables

**Figure 1 biomolecules-15-01173-f001:**
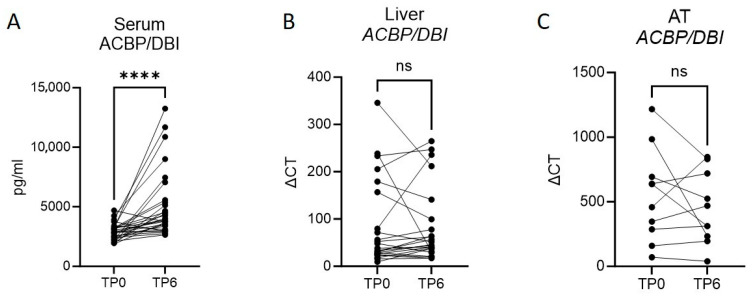
Serum levels of ACBP/DBI increase significantly 6 months after bariatric surgery. (**A**) ACBP/DBI levels in sera of patients at the time point of bariatric surgery and 6 months after surgery, *n* = 29; (**B**) mRNA expression levels in liver tissue samples of *ACBP/DBI* at the time of bariatric surgery and 6 months after surgery, *n* = 23 (**C**); mRNA levels of *ACBP/DBI* in adipose tissue at the time of bariatric surgery and 6 months after surgery, *n* = 10. Statistical significance was assessed using a Wilcoxon matched-pairs signed rank test (**A**,**B**) or a paired student’s *t*-test (**C**). **** *p* ≤ 0.0001; ns not significant.

**Figure 2 biomolecules-15-01173-f002:**
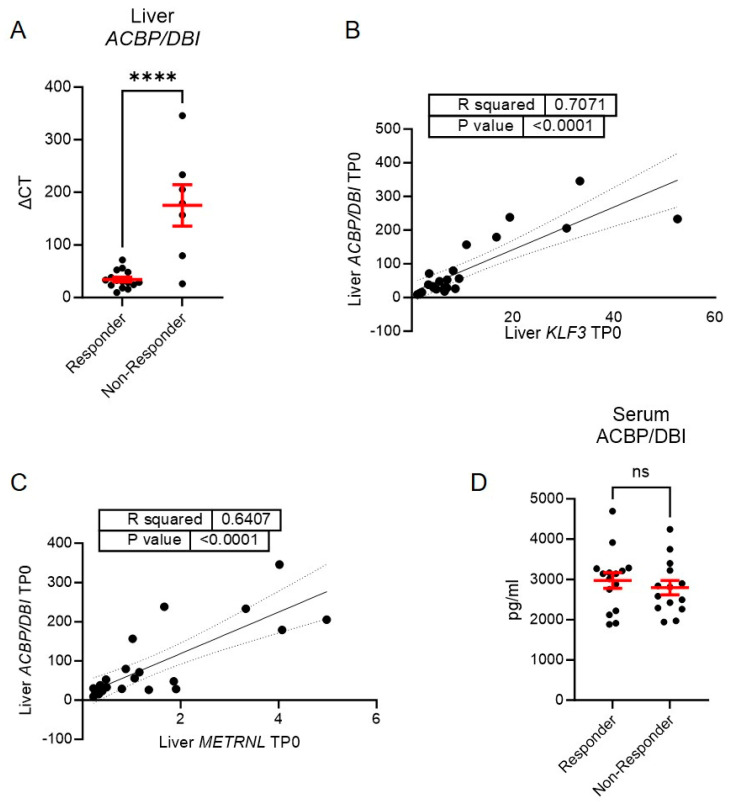
A lower expression of *ACBP/DBI* in liver tissue at time point of operation is observable in patients with a better treatment response. (**A**) mRNA expression levels of *ACBP/DBI* in liver tissue samples at the time of bariatric surgery *n* = 15 responder, *n* = 7 non-responder; (**B**,**C**) Correlation of *ACBP/DBI* mRNA expression levels in liver tissue at time point of bariatric surgery with *KFL3* (**B**) and *METRNL* (**C**) mRNA expression in liver tissue *n* = 23; (**D**) Serum levels of patients undergoing bariatric surgery at the time of surgery, compared between responders (weight loss ≥ 20 kg) and non-responders (weight loss < 20 kg) *n* = 15 responder, *n* = 14 non-responder. Statistical significance was assessed using a simple linear regression to assess if the slope is significantly non-zero (**B**,**C**), or an unpaired student’s *t*-test (**A**,**D**). **** *p* ≤ 0.0001; ns not significant.

**Figure 3 biomolecules-15-01173-f003:**
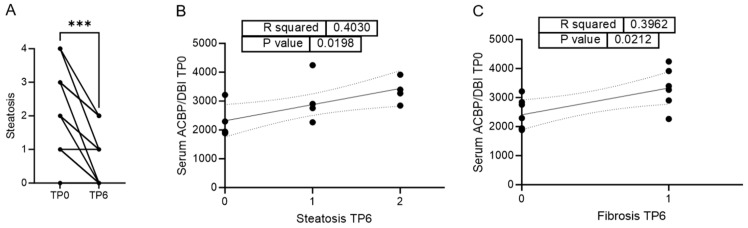
Higher ACBP/DBI serum levels are associated with higher steatosis and fibrosis score 6 months after LAGB. (**A**) Histological scores for steatosis at the time of bariatric surgery (TP0) or six months after surgery (TP6) *n* = 14; (**B**) Correlation between ACBP/DBI serum levels at the time of bariatric surgery and the histological grade of steatosis six months after surgery *n* = 13; (**C**) Correlation between ACBP/DBI serum levels at the time of bariatric surgery and the histological grade of fibrosis six months after surgery *n* = 13. Statistical significance was assessed using a Wilcoxon matched-pairs signed rank (**A**) or a simple linear regression to assess if the slope is significantly non-zero (**B**,**C**). *** *p* ≤ 0.001.

**Figure 4 biomolecules-15-01173-f004:**
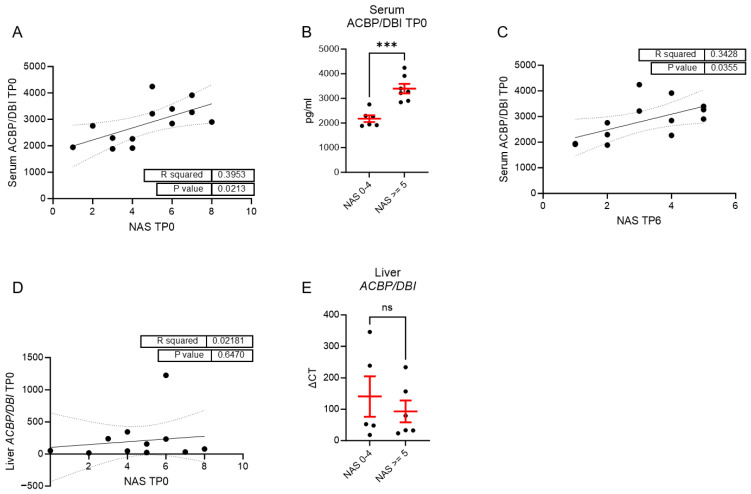
Higher ACBP/DBI serum levels are associated with higher histological NAS score. (**A**) Correlation of NAS and serum ACBP/DBI levels at the time of bariatric surgery *n* = 13; (**B**) comparison of serum ACBP/DBI levels between NAS ≤ 4 and NAS ≥ 5; *n* = 6 NAS ≤ 4, *n* = 7 NAS ≥ 5; (**C**) Correlation between ACBP/DBI serum levels at the time of bariatric surgery and the NAS 6 months after surgery *n* = 13; (**D**) Correlation of NAS and *ACBP/DBI* mRNA levels in liver tissue at the time of bariatric surgery *n* = 12; (**E**) comparison of mRNA liver tissue expression of *ACBP/DBI* between NAS ≤ 4 and NAS ≥ 5; *n* = 5 NAS ≤ 4, *n* = 6 NAS ≥ 5. Statistical significance was assessed using a Mann-Whitney test (**B**,**E**) or a simple linear regression to assess if the slope is significantly non-zero (**A**,**C**,**D**). *** *p* ≤ 0.001; ns not significant.

**Figure 5 biomolecules-15-01173-f005:**
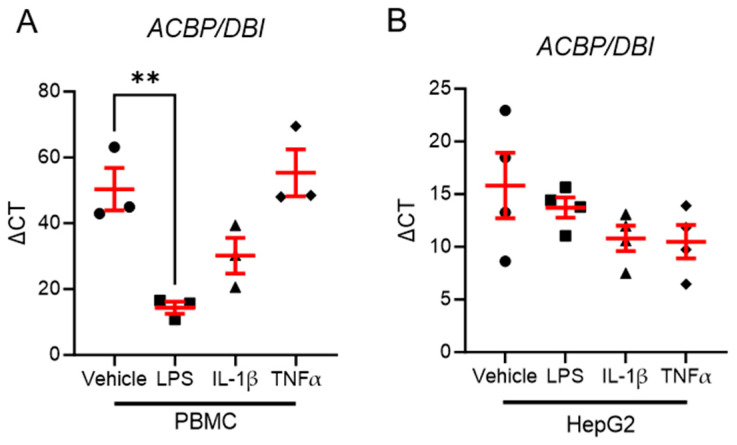
LPS reduces *ACBP/DBI* expression in PBMC of healthy donors but not in an in vitro hepatocyte cell culture model. (**A**,**B**) *ACBP/DBI* expression levels in PBMC of healthy donors (**B**) or the human hepatocyte cell line HepG2 (**B**) stimulated with LPS, IL-1β, TNFα, or vehicle for 24 h, *n* = 3 (**A**) and *n* = 4 (**B**). Statistical significance was assessed with a one-way Anova with Sidak’s multiple comparison. ** *p* ≤ 0.01.

**Table 1 biomolecules-15-01173-t001:** Patient characteristics at time point of surgery (TP0) and six months after surgery (TP6).

Parameter	TP0	TP6	*n*	*p*-Value
Age (years)	37.23 ± 10.86	-	30	-
Sex (F/M)	25/5 (83%/17%)	-	30	-
Weight (kg)	122.3 ± 18.8	102.3 ± 17.1	30	<0.0001
BMI	42.69 ± 3.54	35.8 ± 4.52	30	<0.0001
HOMA-IR	3.5 (2.7–6.8)	2.4 (1.5–3.4)	30	<0.0001
Insulin (µU/mL)	15.30 (11.75–27.53)	11.00 (7.08–14.68)	30	<0.0001
Leptin (ng/mL)	30.11 (26.37–32.31)	17.04 (12.26–25.03)	29	<0.0001
CRP (mg/dL)	0.82 ± 0.46	0.63 ± 0.34	26	0.0081
FIB-4 Score	0.86 ± 0.48	0.84 ± 0.37	24	0.0248

## Data Availability

The original contributions presented in this study are included in the article/Appendix A. Further inquiries can be directed to the corresponding author.

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
