# Peer review of "Hepatic Expression of ACBP Is a Prognostic Marker for Weight Loss After Bariatric Surgery"

_biomolecules, 2025, doi:10.3390/biom15081173_

Round 1
Reviewer 1 Report
Comments and Suggestions for Authors
This study evaluates the effects of metabolic surgery—specifically, laparoscopic adjustable gastric banding—in response to the global challenge of increasing obesity and related cardiometabolic diseases, and reports on the role of **Acyl-CoA Binding Protein (ACBP, also known as diazepam-binding inhibitor)**. ACBP is recognized as a regulator of autophagy (the process by which cells degrade and recycle components) as well as metabolism, and it has recently been identified as an important factor whose blood levels increase during fasting or starvation conditions such as cancer cachexia. Focusing on ACBP is therefore considered highly significant. Furthermore, it is a novel finding that serum ACBP levels significantly increased after surgery, and that preoperative hepatic expression of ACBP predicted the amount of weight loss six months post-surgery. These findings are expected to make a meaningful contribution to future clinical practice.
On the other hand, I would like to point out several issues.
- It has been shown that ACBP increases after surgery, but it is unclear whether this is a result of metabolic improvement or a factor that induces metabolic improvement. Since there is only a correlation and no proof of causality, I would like this point to be discussed.
- Although it has been mentioned that ACBP is involved in autophagy and metabolism, the explanation of how it contributes to weight loss through specific mechanisms is insufficient. Additionally, I would like a more detailed discussion, with concrete examples, regarding its superiority over other biomarkers (e.g., FGF21, leptin, adiponectin, etc.).
- The evaluation period is limited to six months, and the association with sustained weight loss or metabolic markers after one or two years remains unclear. Regarding future clinical applications, it is necessary to discuss whether ACBP would be useful as a long-term biomarker.
Reviewer 2 Report
Comments and Suggestions for Authors
Thank you for the opportunity to review this important manuscript. Here are my comments and suggestions.
In the sentence ''were collected via the routine laboratory and calculated as usually (e.g FIB4). .'' remove one full stop.
The study was conducted 20 years. And some parts of the analysis ''2.2. Quantification of hepatic and adipose tissue ACBP/DBI expression. Expression analysis was performed on samples from a subset of patients (n = 23; 18 females, 5 males) from the original study, where samples were still available.'' Were made probably long after the samples were taken. When were these samples analyzed with this particular method? Aslo, not all patients underwent this analysis.
I understand that the sample is small, but were there any sex differences?
In the discussion, the authors state ''In this study, we show that serum ACBP/DBI levels are significantly increased six months following weight loss induced by bariatric surgery in our cohort. This finding contrasts with a recent study by Bravo-San Pedro et al. (17) , which reported a decrease in ACBP/DBI levels one year after gastric bypass surgery. ''. This should be rephrased and better explained because these two procedures are physiologically and metabolically different. Malnutrition procedure compared to the Restrictive procedure. Perhaps this is why gastric banding is no longer used. The patients regain weight through several mechanisms.
The authors claim ''This observation indicates that the liver expression of ACBP/DBI could act as a biomarker to choose the right treatment for the patient.'' Can they explain this? What could be changed in the management of these patients according to preoperative levels of ACBP?
Round 2
Reviewer 2 Report
Comments and Suggestions for Authors
All queries answered.